# Analysis of Serotonin Transporter Gene *5-HTTLPR* Polymorphism and Its Impact on Personality Traits in a Sample Without Neuropsychiatric or Substance Use Disorders

**DOI:** 10.3390/ijms26083718

**Published:** 2025-04-15

**Authors:** Milena Lachowicz, Aleksandra Suchanecka, Krzysztof Chmielowiec, Agnieszka Boroń, Jolanta Chmielowiec, Katarzyna Prabucka, Monika Rychel, Agnieszka Pedrycz, Remigiusz Recław, Mansur Rahnama-Hezavah, Ewelina Grywalska, Anna Grzywacz

**Affiliations:** 1Department and Clinic of Oncology and Radiotherapy, Medical University of Gdansk, ul. M. Skłodowskiej-Curie 3a, 80-210 Gdansk, Poland; milena.lachowicz@awf.gda.pl; 2Independent Laboratory of Behavioral Genetics and Epigenetics, Pomeranian Medical University in Szczecin, Powstańców Wielkopolskich 72 St., 70-111 Szczecin, Poland; aleksandra.suchanecka@pum.edu.pl (A.S.); katarzyna.prabucka@pum.edu.pl (K.P.); remigiusz.reclaw@pum.edu.pl (R.R.); 3Department of Hygiene and Epidemiology, Collegium Medicum, University of Zielona Góra, 28 Zyty St., 65-046 Zielona Góra, Poland; chmiele@vp.pl (K.C.); chmiele1@o2.pl (J.C.); 4Department of Clinical and Molecular Biochemistry, Pomeranian Medical University in Szczecin, Powstańców Wielkopolskich 72 St., 70-111 Szczecin, Poland; agnieszka.boron@pum.edu.pl (A.B.); monika.rychel@pum.edu.pl (M.R.); 5Faculty of Medicine and Health Sciences, University of Applied Sciences in Tarnow, Mickiewicza 8, 33-100 Tarnów, Poland; apw4@wp.pl; 6Chair and Department of Oral Surgery, Medical University of Lublin, 6 Chodzki Street, 20-093 Lublin, Poland; mansur.rahnama-hezavah@umlub.pl; 7Department of Experimental Immunology, Medical University of Lublin, 4a Chodzki Street, 20-093 Lublin, Poland; ewelina.grywalska@umlub.pl; 8Department of Medical Sciences and Public Health, Gdansk University of Physical Education and Sport, Kazimierza Górskiego 1 St., 80-336 Gdansk, Poland

**Keywords:** personality traits, *5HTTLPR*, serotonin transporter, NEO-FFI

## Abstract

Variations within the serotonin transporter gene, *SLC6A4* (solute carrier family 6 member 4), particularly the *5-HTTLPR* (serotonin-transporter-linked promoter region), have been extensively studied in relation to behavioral and psychological traits. The aim of our study is to examine the relationship between the *5-HTTLPR* polymorphism located in the *SLC6A4* gene and personality traits, as assessed using the NEO-FFI (NEO Five Factor Inventory). The MANOVA model demonstrated a significant overall association, accounting for approximately 8% of the variance in the data (Wilk’s λ = 0.847, F_10,342_ = 2.979, *p* = 0.0013, η^2^ = 0.08). Subsequent ANOVAs revealed statistically significant *5-HTTLPR* polymorphism associations with the Neuroticism (*p* = 0.0018, R^2^ = 0.070), Openness (*p* = 0.0364, R^2^ = 0.037), and Conscientiousness (*p* = 0.0020, R^2^ = 0.068) dimensions. The post-hoc analysis revealed that individuals with the LL genotype obtained significantly lower Neuroticism scores compared to the S/S (*p* = 0.0011) and SL genotype (*p* = 0.0086) carriers. Similarly, individuals with the L/L genotype had lower Openness scores compared to those with SS genotype (*p* = 0.0107). LL and SL genotype carriers had higher Conscientiousness scores compared to those with the SS genotype (*p* = 0.0004 and *p* = 0.0109, respectively). In conclusion, our study provides further data regarding the implications of *5-HTTLPR* polymorphism in the complex genetic architecture of human personality. The observed associations with Neuroticism, Openness, and Conscientiousness, while modest in effect size, contribute to our understanding of how genetic variation at the *SLC6A4* locus may subtly shape individual personality differences.

## 1. Introduction

The concept of personality traits, defined as relatively stable dispositions in cognition, affect, and behaviour [1], is often investigated in relation to various forms of psychopathology, revealing significant phenotypic correlations [2]. Determining the relative contributions of genetic and environmental factors to these observed relationships is a key goal of personality research. Twin studies report heritability estimates of around 40% for personality traits, which suggests a substantial influence of additive genetic effects [3,4,5,6]. Family and adoption studies, which provide another method for disentangling genetic and environmental influences by examining the similarity of traits among biological relatives, including those raised apart, tend to yield somewhat lower heritability estimates, typically around 30% [7,8].

Research suggests that the heritability of personality traits does not substantially differ between males and females [9]. This conclusion is supported by studies comparing the trait resemblance of opposite-sex and same-sex dizygotic twin pairs. For instance, a comprehensive meta-analysis of data from more than 29,000 twin pairs revealed a heritability estimate of 48% for Neuroticism [5], which suggests that the same proportion of variance in Neuroticism is attributable to genetic influences in both men and women. While the relative contribution of genes to personality may be consistent across sex, environmental factors and gene–environment interactions can modulate the expression of these genetic effects [10,11].

A widely accepted framework for understanding personality, the Five-Factor Model [12], proposes that individual differences can be broadly categorised into five core dimensions. These “Big Five” traits encompass Neuroticism (a predisposition to experience negative emotions), Extraversion (marked by an inclination toward social interaction), Openness to experience (characterised by intellectual curiosity and a willingness to try new things), Agreeableness (reflecting qualities like empathy and kindness), and Conscientiousness (exemplified by organisation and self-discipline). Researchers investigating the genetic underpinnings of these traits often utilise the NEO Personality Inventory and its related versions (NEO-PI-R, NEO-FFI) to assess these personality characteristics.

Heritability estimates for personality traits, derived from various questionnaires, typically fall within a similar range, although some variability exists across studies [13,14,15,16,17,18,19,20,21,22,23,24]. The Big Five traits have been shown to exhibit heritability estimates ranging from approximately 31% to 41% [6,25]. The observed variability in these estimates is likely due to factors beyond the traits themselves, including differences in study samples, methodological approaches, and underlying assumptions in twin study designs [8,26,27].

Variations within the serotonin transporter gene, *SLC6A4*, particularly 5-*HTTLPR*, have been extensively studied in relation to behavioural and psychological traits [28,29,30]. This polymorphism, located within the promoter region of *SLC6A4*, influences the expression level of the serotonin transporter (*5-HTT*) [28,30,31]. Specifically, the short (S) allele of the *5-HTTLPR* has been linked to reduced *SLC6A4* transcription and increased vulnerability to various conditions. Compared to individuals homozygous for the long (L) allele, those with the SS or LS genotypes tend to display higher levels of Neuroticism, reduced Agreeableness [28,30], and a greater likelihood of experiencing conditions such as posttraumatic stress disorder, depression [32,33], obsessive–compulsive disorder (OCD) [33], elevated hostility [34], panic disorder [35], and criminal behaviour [36]. Furthermore, the SS and LS genotypes have been associated with a heightened risk of antisocial behaviours, including aggression, impulsivity, and substance use disorders [37,38,39]. An investigation involving the Russian population revealed a higher frequency of the SS genotype in athletes involved in combat sports compared to controls [40,41]. While some initial studies reported associations between the short allele and increased anxiety-related personality traits [28], these findings have not been consistently replicated [42,43,44]. Current research emphasises that the *5-HTTLPR* influences an individual’s sensitivity to environmental influences [45,46]. Carriers of the S allele demonstrate heightened reactivity to environmental stimuli, displaying increased emotional responses [47], stronger seasonal effects on 5-HTT levels [48], and an elevated startle response [49]. The SS genotype, common in European populations [50], is associated with lower *5-HTT* mRNA expression compared to LL. Experiments in mouse models further demonstrate the functional significance of 5-HTT: deletion of *5-HTT* expression significantly increases extracellular serotonin (5-HT) levels, while overexpression leads to a reduction [51,52]. However, the precise neural pathways through which *5-HTTLPR* influences behaviour remain to be fully elucidated. Initial studies proposed that carriers of the S allele showed increased amygdala activation in response to negative stimuli [53,54,55]. This finding, however, has not been consistently supported by subsequent research [56,57]. Moreover, meta-analyses have revealed a weak and/or inconsistent link between the *5-HTTLPR* genotype and amygdala response [56,58], which suggests a need to re-evaluate the role of this brain region in mediating genetic effects.

The present study was designed to examine the relationship between the *5-HTTLPR* polymorphism and personality traits, as assessed using the NEO-FFI. To enhance the interpretability of our findings and minimise potential confounding factors, we implemented specific inclusion criteria. Notably, participants were required to have no history of neuropsychiatric or substance use disorders, and, crucially, they were excluded if they had ever smoked more than one package of cigarettes (twenty cigarettes). This decision was based on the study by Smolka et al. [59], which identified a significant moderating effect of smoking status on the relationship between the *5-HTTLPR* genotype and *5-HTT* expression. The study showed that only in those who smoked did the *5-HTTLPR* genotype not have its usual effects. Furthermore, they also hypothesised that epigenetic processes, such as *SLC6A4* methylation influenced by smoking, might contribute to this effect. By restricting our analyses to individuals with no history of significant tobacco use, we aimed to reduce the likelihood of smoking-related alterations in *5-HTT* expression and its influence on the *5-HTTLPR* and personality. This approach will provide a more reliable assessment of the direct relationship between the *5-HTTLPR* and personality traits.

## 2. Results

Hardy–Weinberg equilibrium (HWE) analysis, including a test for potential ascertainment bias, was conducted for the *5-HTTLPR* polymorphism in the analysed group (Table 1), indicating no significant (*p* = 0.0883) deviation from HWE.

Descriptive statistics for age and NEO-FFI personality dimensions in a sample of 179 individuals are detailed in Table 2. The sample’s average age was approximately 50 years (M = 49.59, SD = 10.93). The NEO-FFI scales exhibited the following means and standard deviations: Neuroticism (3.91 ± 2.19), Extraversion (5.98 ± 2.17), Openness (5.85 ± 2.09), Agreeableness (7.46 ± 2.03), and Conscientiousness (5.89 ± 2.01). Among the NEO-FFI scales, Neuroticism showed the highest relative variability (CV = 55.90%), while Agreeableness demonstrated the lowest (CV = 27.14%).

Moreover, we performed sex-stratified analysis of *5-HTTLPR* variants (χ^2^ = 0.467, *p* = 0.79175) and NEO-FFI traits scores (Neuroticism Z = −1.193, *p* = 0.23291; Extraversion Z = 0.031, *p* = 0.97514; Openness Z = 0.164, *p* = 0.86935; Agreeability Z = 0.665, *p* = 0.506152; Conscientiousness Z = 0.774, *p* = 0.43898).

Furthermore, a correlation analysis was performed between the subjects’ age and the NEO-FFI traits scores (Neuroticism r = −0.061, *p* = 0.42036; Extraversion r = 0.0003, *p* = 0.99654; Openness r = 0.034, *p* = 0.64757; Agreeability r = −0.057, *p* = 0.45164; Conscientiousness r = −0.066, *p* = 0.37995).

The interplay between NEO-FFI trait scales and the *5-HTTLPR* polymorphism was explored using a multivariate analysis of variance (MANOVA) and subsequent series of one-way analyses of variance (ANOVAs). The MANOVA model demonstrated a significant overall association, accounting for approximately 8% of the variance in the data (Wilk’s λ = 0.847, F_10,342_ = 2.979, *p* = 0.0013, η^2^ = 0.08, power (alpha = 0.05) = 0.9798). Presented in Table 3, ANOVAs revealed statistically significant *5-HTTLPR* polymorphism associations with the Neuroticism (F_2,176_ = 6.58, *p* = 0.0018, R^2^ = 0.070), Openness (F_2,176_ = 3.38, *p* = 0.0364, R^2^ = 0.037), and Conscientiousness (F_2,176_ = 6.46, *p* = 0.0020, R^2^ = 0.068) scale scores (Figure 1).

Table 4 presents the results of the post-hoc analysis. Individuals with the LL genotype obtained significantly lower Neuroticism scale scores compared to those with the SS genotype (*p* = 0.0011) or SL genotype (*p* = 0.0086). Similarly, individuals with the LL genotype obtained significantly lower Openness scale scores compared to those with the SS genotype (*p* = 0.0107). On the other hand, individuals with LL or SL genotypes obtained significantly higher Conscientiousness scale scores compared to those with the SS genotype (*p* = 0.0004 and *p* = 0.0109, respectively).

## 3. Discussion

In the present study, we examined the relationship between the *5-HTTLPR* polymorphism and personality traits in a selected sample of volunteers without a history of neuropsychiatric or substance use disorders who have not smoked more than 20 cigarettes in their lives. By restricting our analyses to individuals with no history of significant tobacco use, we aimed to reduce the likelihood of smoking-related alterations in *5-HTT* expression [59] and their influence on the *5-HTTLPR* and personality association. In the studied sample, the mean scores for Extraversion, Openness, and Conscientiousness were average, the Neuroticism mean was low, and Agreeableness was high. Neuroticism showed the highest relative variability, while Agreeableness demonstrated the lowest. The MANOVA model demonstrated a significant overall association between the analysed personality traits and *5-HTTLPR* genotypes. Further analysis revealed statistically significant genotype associations with the Neuroticism, Openness, and Conscientiousness scale scores, with effect sizes from ~4% for Openness and 7% for Neuroticism and Conscientiousness, suggesting limited but significant predictive power. Notably, those with the LL variant obtained lower Neuroticism scores than those with the SS or SL genotypes. The LL genotype might be linked to a somewhat enhanced resilience or lower propensity for experiencing negative emotions in our study population [60]. Similarly, the LL genotype was also associated with reduced scores in Openness compared to the SS genotype, which suggests a possible inclination towards more conventional or less exploratory tendencies in those with this genotype [60]. Conversely, regarding Conscientiousness, the LL and SL genotype carriers obtained higher scores than those with the SS variant. This might indicate a propensity toward organisation, responsibility, and goal-directed behaviour in individuals carrying at least one L allele [12]. Moreover, we performed a sex-stratified analysis of *5-HTTLPR* variants and NEO-FFI trait scores, finding both analyses insignificant. We also analysed the influence of age on the NEO-FFI trait scores, but the analysis did not yield significant results, which suggests that the observed associations may be relatively consistent across different sexes and age groups within our study population. Our results align with meta-analytic findings by Munafò et al. [60], who analysed the moderating effect of sex on the association between the *5-HTTLPR* S allele and Neuroticism in a sample of mainly Caucasian origin. Our study associated the L allele with higher Conscientiousness scores. The role of serotonin in impulse control [61] implies that the higher serotonin levels facilitated by the L allele could contribute to better impulse control, a key facet of conscientiousness. Enhanced executive function, also linked to serotonin and prefrontal cortex activity [62], could further support goal-directed behaviour and planning, which are characteristics of conscientious individuals [12].

The association of the S allele with both Openness and Neuroticism may be linked to psychological flexibility (PF), which is also positively associated with the S allele [63]. PF is understood as the capacity to fully engage with the present experience without judgement while either maintaining or adapting behaviour to align with personal values [64], and it is increasingly recognised as a crucial element in adaptive human functioning. This construct, in essence, reflects an individual’s ability to effectively respond to both external environmental cues and internal experiences in a manner that promotes goal attainment [65]. Some researchers suggest that psychological flexibility may have played a significant role in human evolution [66]. This capacity is considered fundamental to overall psychological well-being [67,68]. Changes in psychological flexibility were analysed [63] during exposure-based cognitive behavioural treatment. Individuals with the S allele experienced almost twice the improvement in PF than those with the L allele. Previous studies show that the effects of the short allele of the *5-HTTLPR* polymorphism depend on context [46]. This observation is consistent with an evolutionary framework. The ability of short allele carriers to more easily increase PF offers a potential explanation for its persistence in the population. This supports the classification of this polymorphism as a plasticity variant [45], not merely a vulnerability [69]. The mechanisms by which the S allele might contribute to gains in psychological flexibility remain under investigation. Psychological flexibility is related to executive functions mediated by the frontal cortex [70]. Studies suggest that individuals carrying the short allele for the *5-HTTLPR* may exhibit superior attention and working memory capabilities [71,72]. This raises the possibility that the S allele enhances frontal cortex function, potentially boosting PF. However, this benefit might be accompanied by heightened sensitivity and reactivity to adverse environmental conditions and negative emotions. While the S allele may predispose individuals to higher Neuroticism, it may also enhance Openness to new experiences, thereby influencing the capacity for psychological flexibility.

This study has several limitations. The sample was comprised only of individuals of Caucasian origin, limiting generalisability. We used only basic demographic, genetic, and psychometric data without adjusting for sociodemographic factors, stressors, or life events. However, the participants met very specific inclusion criteria, including exclusion of any history of neuropsychiatric illness, substance dependencies, or nicotine use. We also only analysed the 5-*HTTLRP* polymorphic site without expression studies. Finally, the findings rely on a single sample, lacking replication or internal validation due to the modest sample size.

In conclusion, the present study examines the influence of the *5-HTTLPR* polymorphism on personality traits (NEO-FFI) in a sample of participants without a history of neuropsychiatric or substance use disorders, as well as limited lifetime smoking. After controlling for age and sex, results from MANOVA indicated a significant overall association between the *5-HTTLPR* genotype and personality profiles. Specifically, the LL genotype was associated with lower Neuroticism and Openness scores, whereas the LL and SL genotypes were associated with elevated Conscientiousness scores. The strength of these associations suggests a measurable but modest relation of the *5-HTTLPR* genotype with these personality dimensions.

## 4. Materials and Methods

### 4.1. Participants

The study group comprised 179 volunteers, 139 of whom were females (mean age = 49.31, SD = 10.57) and 40 who were males (mean age = 49.95, SD = 11.70). The participants are Polish of Caucasian origin. The individuals were screened to exclude any history of neuropsychiatric illness, including substance dependency and smoking status. The screening was performed by a psychiatrist using the Mini International Neuropsychiatric Interview (MINI) and NEO Five-Factor Inventory to assess the personality traits scores.

The Bioethics Committee of the Pomeranian Medical University in Szczecin approved the study (KB-0012/164/17-A). All participants submitted their written informed consent to volunteer in the study. The study was conducted in the Independent Laboratory of Behavioral Genetics and Epigenetics.

### 4.2. Psychometric Measures

The MINI-International Neuropsychiatric Interview is a structured diagnostic interview designed to evaluate individuals’ diagnoses according to the DSM-IV and ICD-10 criteria.

The NEO Five-Factor Inventory assesses six components for each of the five traits–neuroticism (anxiety, hostility, depression, self-awareness, impulsivity, susceptibility to stress), extroversion (warmth, sociability, assertiveness, activity, emotion seeking, positive emotions), openness to experience (fantasy, aesthetics, feelings, actions, ideas, values), agreeableness (trust, straightforwardness, altruism, compliance, modesty, tenderness), and conscientiousness (competence, order, duty, striving for achievements, self-discipline, consideration).

The results of NEO-FFI were reported as sten scores. The transition of the raw score to the sten scale was conducted according to the Polish standards for adults, as follows: 1–2 corresponding to very low results, 3–4 to low results, 5–6 to average results, 7–8 to high results, and 9–10 to very high results.

### 4.3. Genotyping

Genomic DNA was isolated from blood leukocytes using the QIAapm^®^ DNA Mini Kit (Qiagen, Hilden, Germany); all samples were sufficient for amplification. The VNTR polymorphism (ins/del 44bp; *5-HTTLPR*) in the *SLC6A4* gene were identified with polymerase chain reaction (PCR), using specific pairs of primers: 5′-GGC GTT GCC GCT CTG AAT GC-3′ (forward) and 5′-GAG GGA CTG AGC TGG ACA ACC AC-3′ (reverse). The PCR products were separated by electrophoresis in 2% agarose gel and visualised under an ultraviolet (UV) gel imaging system Gbox (SYNGENE, Cambridge, United Kingdom). Imaging analysis detected 528 bp and 484 bp products, the 484 bp being the “S” or “short” variant, while the 528 bp is the “L” or “long” variant. The PCR products were compared with pUC19 DNA/*MspI* (*HpaII*) Marker (Thermo Fisher Scientific, Waltham, MA, USA) (Figure 2).

For internal control, 10% of randomly selected samples were re-analysed by PCR and electrophoresis-based genotyping. The results were 100% confirmed.

### 4.4. Statistical Analysis

A concordance between the genotype distribution and Hardy–Weinberg equilibrium was tested using the HWE software (https://wpcalc.com/en/equilibrium-hardy-weinberg/ (accessed on 11 November 2024)). The sex-stratified analysis of the NEO Five-Factor Inventory trait scores was performed using the Mann–Whitney U test. The chi-square test was employed to conduct a sex-stratified association analysis of the *5-HTTLPR* genotypes. The correlation between the subjects’ age and the NEO-FFI trait scores was calculated using Pearson linear correlation. The relationships between *5-HTTLPR* variants and the NEO Five-Factor Inventory trait scores were analysed using a multivariate analysis of variance (MANOVA). The condition of homogeneity of variance was met (Levene test *p* > 0.05).

All computations were executed using STATISTICA 13 (Tibco Software Inc, Palo Alto, CA, USA) for Windows (Microsoft Corporation, Redmond, WA, USA).

## 5. Conclusions

In summary, our study provides further data regarding the *5-HTTLPR* polymorphism’s role in the complex genetic architecture of human personality. The observed associations with Neuroticism, Openness, and Conscientiousness, while modest in effect size, contribute to our understanding of how genetic variation at the *SLC6A4* locus may subtly shape individual personality differences.

## Figures and Tables

**Figure 1 ijms-26-03718-f001:**
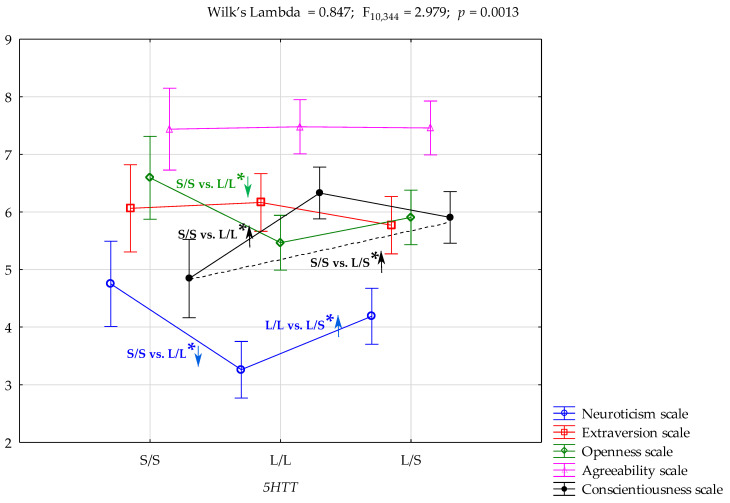
Interaction of the NEO-FFI scales scores and *5-HTTLPR* polymorphism in a sample without neuropsychiatric or substance use disorders. *—significant result. ↓ and ↑ indicate lower or higher score of the trait relative to the *5-HTTLPR* variant, respectively.

**Figure 2 ijms-26-03718-f002:**
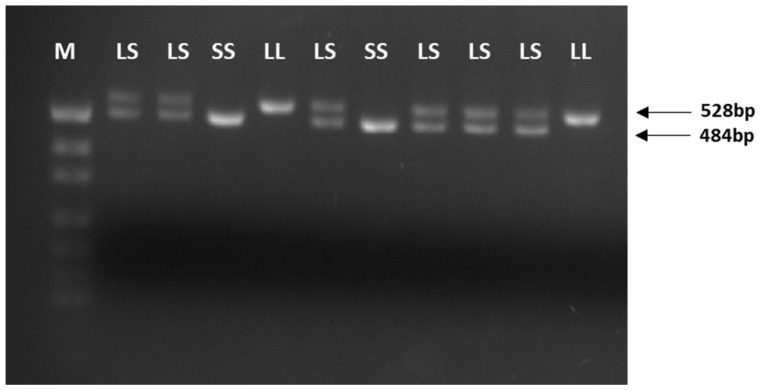
The visualisation of the *5-HTTLPR* polymorphism PCR products. Lanes: M—pUC19 DNA/*MspI* (*HpaII*) Marker. LS—heterozygote Long/Short (528 bp/484 bp). LL—homozygote Long/Long (528 bp/528 bp). SS—homozygote Short/Short (484 bp/484 bp).

**Table 1 ijms-26-03718-t001:** Hardy–Weinberg’s law analysis for the *5-HTTLPR* polymorphism in a sample without neuropsychiatric or substance use disorders.

Hardy–Weinberg Equilibrium Including Analysis for Ascertainment Bias		Observed (Expected)	Allele Freq	χ^2^ (*p* Value)
Subjects n = 179	LL	73 (67.6)	p (L) = 0.71 q (S) = 0.39	2.9055 (0.0883)
	LS	74 (84.8)		
	SS	32 (26.6)		

L—long; S—short; *p*—statistical significance χ^2^ test.

**Table 2 ijms-26-03718-t002:** Age and the NEO-FFI scores in a sample without neuropsychiatric or substance use disorders.

Characteristic	Age	Neuroticism Scale	Extraversion Scale	Openness Scale	Agreeability Scale	Conscientiousness Scale
Subjects (n)	179	179	179	179	179	179
Mean (M)	49.59	3.91	5.98	5.85	7.46	5.89
Standard deviation (SD)	10.93	2.19	2.17	2.09	2.03	2.01
Coefficient of variation% (V)	22.04	55.90	36.20	35.72	27.14	34.10

**Table 3 ijms-26-03718-t003:** ANOVA for the *5-HTTLPR* genotypes and the NEO Five-Factor Inventory scales in a sample without neuropsychiatric or substance use disorders.

NEO-FFI Scale	LL	LS	SS	R^2^	F_2,176_	*p*
	(n = 73)	(n = 74)	(n = 32)			
Neuroticism	3.26 (1.82)	4.18 (2.42)	4.75 (2.02)	0.070	6.58	0.0018 *
M (SD)						
Extraversion	6.16 (2.15)	5.77 (2.20)	6.06 (2.14)	0.007	0.63	0.5328
M (SD)						
Openness	5.47 (2.06)	5.91 (2.01)	6.59 (2.18)	0.037	3.38	0.0364 *
M (SD)						
Agreeability	7.48 (1.95)	7.46 (2.20)	7.44 (1.83)	0.0001	0.005	0.995
M (SD)						
Conscientiousness M (SD)	6.33 (2.06)	5.91 (2.04)	4.84 (1.42)	0.068	6.46	0.0020 *

*—significant result; M ± SD—mean ± standard deviation; R^2^—Coefficients of determination; F_2,176_—Fisher’s test.

**Table 4 ijms-26-03718-t004:** Post-hoc (Least Significant Difference) analysis of interactions between the NEO Five-Factor Inventory Neuroticism, Openness, and Conscientiousness scores in a sample without neuropsychiatric or substance use disorders.

*5-HTTLPR* Genotypes			
Neuroticism			
	SS	LL	LS
	M = 4.75	M = 3.26	M = 4.18
SS		0.0011 *↓	0.2130
LL			0.0086 *↑
LS			
Openness			
	SS	LL	LS
	M = 6.59	M = 5.47	M = 5.91
SS		0.0107 *↓	0.1164
LL			0.1978
LS			
Conscientiousness			
	SS	LL	LS
	M = 4.84	M = 6.33	M = 5.91
SS		0.0004 *↑	0.0109 *↑
LL			0.1897
LS			

*—significant statistical differences, M—mean, ↓ and ↑ indicate lower or higher score of the trait relative to the *5HTTLPR* variant, respectively.

## Data Availability

Data is contained within the article.

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
