# Peer review of "Analysis of Serotonin Transporter Gene 5-HTTLPR Polymorphism and Its Impact on Personality Traits in a Sample Without Neuropsychiatric or Substance Use Disorders"

_ijms, 2025, doi:10.3390/ijms26083718_

Round 1
Reviewer 1 Report
Comments and Suggestions for Authors
In this paper, the authors examine correlations between two polymorphisms of the 5-HTT gene and several personality traits, as assessed by the NEO five factor inventory, in a group of 179 individuals with no identified psychiatric disorders or addictions. Some correlations were identified and there may be some value to this assessment of “normal” individuals. Unfortunately, the paper is very poorly written, and even beyond the poor English usage, it is not possible to compare the authors’ results with others, most of which are 20 years old. The paper needs to be entirely re-written. Just a few major comments are listed below, out of many, that would need to be addressed in a possible revision.
- A more comprehensive description of the ethnicity and other traits of the subjects is needed.
- While the study included both males and females, sex-specific data did not seem to be presented.
- The references are completely outdated with almost all at least 20 years old.
- The authors, even in just the abstract begin with a sentence fragment, describe what was done, then say the study “will analyze” (line 11), then repeats what was done again and, finally summarizes the results.
- More discussion of the polymorphism results in comparison to the composite traits measured in each of the five NEO classifications would be valuable.
Minor comments.
- Line 36, there seem to be extra words
- The use of “never smoked”, apparently meaning lacking addiction or psychiatric disease needs to be changed.
Paper needs to be re-written and results put in context of current research.
Author Response
Answer to reviewer
We sincerely thank you for all your comments. The location of all changes is below, which can be additionally seen in the changes tracking panel.
Comments and Suggestions for Authors
In this paper, the authors examine correlations between two polymorphisms of the 5-HTT gene and several personality traits, as assessed by the NEO five factor inventory, in a group of 179 individuals with no identified psychiatric disorders or addictions. Some correlations were identified and there may be some value to this assessment of “normal” individuals. Unfortunately, the paper is very poorly written, and even beyond the poor English usage, it is not possible to compare the authors’ results with others, most of which are 20 years old. The paper needs to be entirely re-written. Just a few major comments are listed below, out of many, that would need to be addressed in a possible revision.
- A more comprehensive description of the ethnicity and other traits of the subjects is needed.
Thank you for this suggestion. The description in the Material and Methods section (lines 281-286) was added and included in the limitations (lines 262-266).
- While the study included both males and females, sex-specific data did not seem to be presented.
Thank you for pointing this out. The sex-stratified analysis has been added to the Results section (lines 147-150).
- The references are completely outdated with almost all at least 20 years old.
Thank you for pointing this out. The references have been updated.
- The authors, even in just the abstract begin with a sentence fragment, describe what was done, then say the study “will analyze” (line 11), then repeats what was done again and, finally summarizes the results.
Thank you for pointing this out. The Abstract has been rewritten.
- It would be valuable to discuss the polymorphism results in relation to the composite traits measured in each of the five NEO classifications more extensively.
Thank you for this suggestion. The information has been added and discussed in the Introduction and Discussion sections.
Minor comments.
Line 36, there seem to be extra words
Thank you for pointing this out. The editing mistake has been corrected.
The use of “never smoked”, apparently meaning lacking addiction or psychiatric disease needs to be changed.
Thank you for pointing this out. The description of the sample has been updated in the entire paper.
Comments on the Quality of English Language
Paper needs to be re-written and results put in context of current research.
Thank you for this suggestion. The entire paper has been rewritten and updated.
Reviewer 2 Report
Comments and Suggestions for Authors
This manuscript addresses a well-studied but still relevant topic: the potential association between 5-HTTLPR polymorphism in the serotonin transporter gene and personality traits, with the unique angle of focusing exclusively on healthy, never-smoking individuals. The authors employed well-established psychometric tools (NEO-FFI, MINI) and applied reasonable genetic and statistical methodologies (PCR genotyping and MANOVA). The findings specifically linking the L/L genotype to lower Neuroticism and Openness but higher Conscientiousness are intriguing and worth exploring further. While the manuscript is labeled as an “Article,” its current structure and length make it feel more like a Short Report or Brief Communication.
The study is generally well-written and structured, and it contributes to the ongoing discourse about gene-personality interactions in healthy populations. However, there are a few methodological, interpretative should be addressed before the manuscript can be considered for publication.
Major Comments
1. The manuscript emphasizes the inclusion of "never smokers," yet it lacks a strong rationale as to why smoking status was used as a key selection criterion. While it’s valid to control for lifestyle factors, this choice would benefit from a clearer justification, does nicotine use have a known or hypothesized interaction with serotonergic pathways in the context of personality?
2. While 179 subjects is reasonable, a more detailed discussion of statistical power is needed. Specifically, is the study powered to detect genotype effects across all five NEO traits? Given the modest effect sizes, a power analysis should be included or discussed.
3. The discussion briefly references gene-environment interaction models (e.g., stress reactivity), but the present study doesn't assess environmental stressors or life events. While it's fine to focus on genotype-personality links, the conclusions should avoid over-interpreting the role of 5-HTTLPR without GxE context.
4. While PCR and gel-based genotyping were used, it would be helpful to include some quality control details: were a subset of samples replicated to confirm genotype accuracy? Was there any discordance?
5. The manuscript reports statistically significant associations but should be cautious not to overstate the biological significance of these findings. The effect sizes (R² values) are quite small (~3-7%), which suggests limited predictive power. This should be clearly acknowledged.
6. The sample includes 139 females and only 40 males, yet the analysis does not account for sex as a covariate or stratify the data. Since both personality traits and serotonergic function can be influenced by sex differences, this imbalance could confound the findings. The authors should either control for sex statistically or provide a justification for not doing so.
7. The participants' mean age is around 49.6 years, but the age range and its potential influence on personality traits or gene expression are not adequately addressed. Age can interact with both genetics and personality development. The authors should discuss whether age was controlled for or examined as a potential covariate.
8. The study relies on a single sample without any form of internal validation (e.g., split-sample) or replication in an independent cohort. Although the sample size is modest, this should be acknowledged as a limitation, particularly given the subtle effect sizes typically observed in candidate gene studies.
Minor Comments
9. Figure 1 could benefit from clearer labeling (e.g., adding p-values or stars to indicate significance directly on the plot).
10. Post-hoc results (Table 4) are informative but may be more readable as a matrix with arrows or a heatmap to quickly show direction and magnitude of effects.
11. The discussion meanders a bit and includes tangents (e.g., dopamine receptor polymorphisms) that aren't directly connected to the main results. Trimming this section to stay focused on the 5-HTT findings would improve clarity.
Comments on the Quality of English LanguageI recommend that the manuscript undergo thorough English editing by a native speaker or professional language editing service to enhance clarity, improve flow, and ensure the scientific message is effectively communicated.
Author Response
Answer to reviewer 2
We sincerely thank you for all your comments. The location of all changes is below; it can also be seen in the changes tracking panel.
This manuscript addresses a well-studied but still relevant topic: the potential association between 5-HTTLPR polymorphism in the serotonin transporter gene and personality traits, with the unique angle of focusing exclusively on healthy, never-smoking individuals. The authors employed well-established psychometric tools (NEO-FFI, MINI) and applied reasonable genetic and statistical methodologies (PCR genotyping and MANOVA). The findings specifically linking the L/L genotype to lower Neuroticism and Openness but higher Conscientiousness are intriguing and worth exploring further. While the manuscript is labeled as an “Article,” its current structure and length make it feel more like a Short Report or Brief Communication.
The study is generally well-written and structured, and it contributes to the ongoing discourse about gene-personality interactions in healthy populations. However, there are a few methodological, interpretative should be addressed before the manuscript can be considered for publication.
Major Comments
- The manuscript emphasizes the inclusion of "never smokers," yet it lacks a strong rationale as to why smoking status was used as a key selection criterion. While it’s valid to control for lifestyle factors, this choice would benefit from a clearer justification, does nicotine use have a known or hypothesized interaction with serotonergic pathways in the context of personality?
Thank you for this question. The sample description has been updated throughout the paper, and the rationale for excluding nicotine users from the study has been added (lines 115-129).
- While 179 subjects is reasonable, a more detailed discussion of statistical power is needed. Specifically, is the study powered to detect genotype effects across all five NEO traits? Given the modest effect sizes, a power analysis should be included or discussed.
Thank you for this question. Our study was sufficiently powered, as you can see from the graphic representation of the power calculation below. The information has been added to the Results section as well (line 159)
- The discussion briefly references gene-environment interaction models (e.g., stress reactivity), but the present study doesn't assess environmental stressors or life events. While it's fine to focus on genotype-personality links, the conclusions should avoid over-interpreting the role of 5-HTTLPR without GxE context.
Thank you for pointing this out. The Conclusion section has been updated to avoid overinterpretation, and information regarding possible GxE has been added to the limitation (lines 263-266)
- While PCR and gel-based genotyping were used, it would be helpful to include some quality control details: were a subset of samples replicated to confirm genotype accuracy? Was there any discordance?
Thank you for this question. Quality control has been performed according to our lab’s SOPs. The information has been added to the Material and Methods section (317-318)
- The manuscript reports statistically significant associations but should be cautious not to overstate the biological significance of these findings. The effect sizes (R² values) are quite small (~3-7%), which suggests limited predictive power. This should be clearly acknowledged.
Thank you for this suggestion. The magnitude of effect sizes is clearly stated and acknowledged in the Discussion and Conclusion sections.
- The sample includes 139 females and only 40 males, yet the analysis does not account for sex as a covariate or stratify the data. Since both personality traits and serotonergic function can be influenced by sex differences, this imbalance could confound the findings. The authors should either control for sex statistically or provide a justification for not doing so.
Thank you for pointing this out. The sex-stratified analysis has been added to the Results section (lines 147-150).
- The participants' mean age is around 49.6 years, but the age range and its potential influence on personality traits or gene expression are not adequately addressed. Age can interact with both genetics and personality development. The authors should discuss whether age was controlled for or examined as a potential covariate.
Thank you for pointing this out. The correlation analysis of NEO scores and age has been added to the Results section (lines 147-150).
- The study relies on a single sample without any form of internal validation (e.g., split-sample) or replication in an independent cohort. Although the sample size is modest, this should be acknowledged as a limitation, particularly given the subtle effect sizes typically observed in candidate gene studies.
Thank you for pointing this out. The information has been added to the limitation (lines 267-269)
Minor Comments
- Figure 1 could benefit from clearer labeling (e.g., adding p-values or stars to indicate significance directly on the plot).
Thank you for this suggestion. The labels have been updated.
- Post-hoc results (Table 4) are informative but may be more readable as a matrix with arrows or a heatmap to quickly show direction and magnitude of effects.
Thank you for this suggestion. The table has been updated.
- The discussion meanders a bit and includes tangents (e.g., dopamine receptor polymorphisms) that aren't directly connected to the main results. Trimming this section to stay focused on the 5-HTT findings would improve clarity.
Thank you for pointing this out. The Discussion section has been rewritten to focus on 5-HTT, serotonin, and personality implications.
Comments on the Quality of English Language
I recommend that the manuscript undergo thorough English editing by a native speaker or professional language editing service to enhance clarity, improve flow, and ensure the scientific message is effectively communicated.
Thank you for pointing this out. The entire paper has been rewritten and verified by a native speaker.

Round 2
Reviewer 1 Report
Comments and Suggestions for Authors
In this reviewer's opinion, the authors have submitted a substantially improved manuscript. As they note, three personality traits were associated with the 5-HTTLPR polymorphisms. While these traits had modest effect sizes, the work was carefully perform, appropriately analyzed and presented with far greater clarity than in the original submission.
Reviewer 2 Report
Comments and Suggestions for Authors
This revised manuscript has been improved by addressing the reviewer’s comments. However, although it is currently labeled as an “Article,” I still believe its structure and length are more appropriate for a Short Report or Brief Communication.